# *Heliotropium procubens* Mill: Taxonomic Significance and Characterization of Phenolic Compounds via UHPLC–HRMS- In Vitro Antioxidant and Enzyme Inhibitory Activities

**DOI:** 10.3390/molecules28031008

**Published:** 2023-01-19

**Authors:** Kalliopi-Maria Ozntamar-Pouloglou, Antigoni Cheilari, Gokhan Zengin, Konstantia Graikou, Christos Ganos, George-Albert Karikas, Ioanna Chinou

**Affiliations:** 1Laboratory of Pharmacognosy & Chemistry of Natural Products, Faculty of Pharmacy, National and Kapodistrian University of Athens, Panepistimiopolis, 15771 Zografou, Greece; 2Department of Biology, Science Faculty, Selcuk University, Konya 42130, Turkey; 3Department of Biomedical Sciences, University of West Attica, 12243 Egaleo, Greece

**Keywords:** Boraginaceae, *Heliotropium procumbens*, phenolic compounds, antioxidant activity, enzyme inhibitory activity

## Abstract

The aim of the present study was the phytochemical analysis of the aerial parts of *Heliotropium procumbens* Mill., a herb from Boraginaceae plant family not previously studied. The methanol (ME) and aqueous extracts (WE) of the aerial parts were assayed for their total phenolic and flavonoid content and antioxidant properties, using free radical scavenging (DPPH, ABTS), reducing power (FRAP, CUPRAC), phosphomolybdenum and metal chelating assays. The extracts displayed considerable free radical scavenging activity against DPPH and ABTS radicals, with potential values of 46.88 and 68.31 mg TE/g extract for ME, and 93.43 and 131.48 mg TE/g extract for WE, respectively. Key clinical enzymes involved in neurodegenerative diseases AChE and BChE, diabetes (α-amylase and α-glucosidase) and skin whitening (tyrosinase) were also assayed. The phytochemical profile of the studied species was determined through UHPLC–HRMS, whereby 26 secondary metabolites were identified, three of which (luteolin-7-glucoside, lithospermic and rosmarinic acids) were isolated and structurally determined by NMR spectral means. *H. procubens* was found to harbor bioactive metabolites and could, hence, serve as a source of biological activities which could be further explored and exploited for potential applications.

## 1. Introduction

The Boraginaceae plant family comprises 156 genera and 2650 species of herbs, shrubs, and trees, with a worldwide distribution, occurring mainly in Europe, Asia and North America [1]. *Heliotropium*, *Arnebia*, *Martensia*, *Cordia*, and *Trichedesma* are the main genera of this family, with therapeutic effects mostly based on the presence of phenolic metabolites [2,3,4]. Moreover, almost all genera of the family Boraginaceae produce and serve as a natural source of pyrrolizidine alkaloids, which are related with serious health problems [4,5]. 

*Heliotropium* (Boraginaceae) is a genus of herbs, distributed in tropical and temperate regions of the world comprising approximately 300 species commonly known as heliotropes. The name “heliotrope” derives from the fact that these plants turn their leaves to the direction of the sun and, as a result, the name includes the word “helios”, a Greek word for “sun”, and “tropein” which means “to turn”. Heliotropes [5] also exhibit several therapeutic properties, such as antimicrobial, anti-inflammatory, analgesic, healing, etc., and they have been used in folk medicine against rheumatism, menstrual dysfunction, billiary disorders, and noxious bites [6,7,8]. The potential active secondary metabolites from the *Heliotropium* species comprise phenolic compounds (phenolic acids, flavonoids, quinones: alkannins–shikonins), terpenoids and pyrrolizidine alkaloids (PAs).

This study has been focused on the species *Heliotropium procumbens* Mill., a herb with white flowers, native to America, as well as East to West Indies, is used by locals in animal feed, as it is considered to exhibit a high nutritional value with a notably high protein content [9]. *H. procumbens* occurs in the USA Gulf Coastal Region (Texas, Mississippi, and Louisiana), and it has been questioned whether it is a native or adventive species in the area. After comparing the time and location of the various Texan specimens assembled over the past 150 years, it has been concluded that *H. procumbens* is native to the area under consideration [10]. 

In the framework of our research on Boraginaceae plants [3,4,11,12,13], we herein present the phytochemical study on the aerial parts of *H. procumbens* from Panama in order to evaluate its chemical composition, the total phenolic and flavonoid content, antioxidant activity, inhibitory potentials against key clinical enzymes involved in neurodegenerative diseases (cholinesterases AChE, BChE), in skin whitening (tyrosinase), and in diabetes (α-amylase, α-glucosidase). 

## 2. Results

### 2.1. Phytochemical Analysis

Through our chemical analysis of secondary metabolites, characteristics of the *Heliotropium* genus, as well as of the whole plant family of Boraginaceae, have been identified and further examined. Several among them contributed to the exerted biological properties, such as the caffeic derivatives, rosmarinic acid (RA), and lithospermic acid together with the flavone luteolin-7-O-glucoside [14]. All three of them have been isolated and structurally determined through NMR.

Furthermore, through UHPLC–HRMS, 26 secondary metabolites have been identified (Figure 1, Table 1, Appendix A). 

### 2.2. Total Phenolic (TPC)/Flavonoid (TFC) Content and Antioxidant Activity Results 

The antioxidative capacity of the aerial parts from *H. procumbens* extracts was determined using different assays (DPPH•, ABTS•+, CUPRAC, FRAP), phosphomolybdenum and ferrous ion chelating test, as well as the TPC and TFC which are correlated with the antioxidant activity. The results (Table 2) show a high TPC (32.20 mg GAE/g for ME and 53.47 mg GAE/g for WE) and a lower TFC (0.70 mg REs/g extract for ME and 12.39 mg REs/g extract for WE), assuming that this plant is a rich source of phenolics, which is in accordance with the presented phytochemical profile.

The DPPH and ABTS assays on the *H. procumbens* methanolic extract showed good inhibition of the free radicals (46.88 ± 0.64 inhibition for the DPPH assay and 68.31 ± 0.69 for the ABTS assay) and, therefore, a significant antioxidative profile, while the water extract showed a higher scavenging capacity (93.43 ± 0.11 for the DPPH assay and 131.48 ± 6.66 for the ABTS assay). 

### 2.3. Enzyme Inhibitory Activity

The results obtained from enzyme inhibitory assays from *H. procumbens* extracts are summarized in Table 3. Regarding the acetylocholinesterase inhibition, the methanolic and aqueous extracts did not exhibit any activity (0.48 ± 0.04 mg GALAEs/g extract for WE), while no inhibition was detected in the case of the butyrylcholinesterases assay (1.17 ± 0.25 mg GALAEs/g extract for ME).

Regarding the inhibitors, which control glucose level in blood, the assayed extracts exhibited higher α-glucosidase inhibitory activity (1.97 ± 0.14 mmol ACAE/g extract ME and 2.08 ± 0.01 mmol ACAE/g extract WE) compared to α-amylase inhibitory activity (0.22 ± 0.01 mmol ACAE/g extract ME and 0.07 ± 0.01 mmol ACAE/g extract WE) but both are not valuable.

In addition, ME showed higher inhibitory capacity against tyrosinase, an enzyme involved in skin whitening, (25.05 ± 0.17 mg KAE/g extract) in comparison with the WE (8.82 ± 0.80 mg KAE/g extract).

## 3. Discussion

From the studied *H. procumbens*, 26 secondary metabolites have been presented so far. Several coumaric and caffeic acid derivatives have been revealed, previously identified in other *Heliotropium* species as *H. lasiocarpium*, *H. suaveolens* [17] and *H. strigosum* [18], and they have been strongly related with the plant’s antioxidant properties. Syringic acid has been also previously identified in *H. curassavicum* [19] and *H. strigosum* [18]. Isobergapten, among coumarins, has been identified previously in several heliotropes, among which are *H. lasiocarpium, H. suaveolens* and *H. crispum* [17]. Methyl-catechin has never been identified in the genus, while catechin has been found in *H. strigosum* [27], *H. crispum* [28] and *H. curassavicum* [19]. Dihydroxy methoxy benzoic acid (pyruvic acid) has been previously identified in *H. crispum* through UHPLC-Q-TOF-MS together with other benzoic acids’ derivatives and/or substituted hydroxy-methoxy benzoic acids [17]. Vanillic acid has been previously found in *H. thermophilum* [29] and *H. strigosum* [18]. Ferulic acid is a coumaric derivative previously reported as either *iso*, *cis* or *trans*-ferulic acid in *H. crispum* [17] and *H. strigosum* [18], not easily distinguished due to common fragmentation between the isomers. Among the most abundant metabolites in the studied heliotrope species were the caffeic acid derivatives, RA and lithospermic acid, together with the flavone luteolin-7-glucoside which were identified but also isolated and structurally determined.

The identified by UHPLC–HRMS caffeic derivatives in this study were caffeic acid itself, caffeoyl hexoside, RA and its methyl ester, lithospermic acid and salvianolic acid B. 

RA, a polyphenolic compound, is structurally an ester of caffeic acid and is derived from hydroxycinnamic acid, widespread and well-known for its multitude of bioactivities, such as antioxidant, antiviral, antimicrobial and anti-inflammatory. It possesses mainly neuroprotective, anti-acetyl cholinesterase and hepatoprotective properties, and is rapidly eliminated from human and rat blood circulation after per os administration, metabolized predominantly to caffeic, coumaric and ferulic acids. According to the results of previous phytochemical studies, RA is considered as a chemotaxonomic marker among Boraginaceae taxa [11,12,13,14], while it has been also isolated from *H. foertherianum* [30] and *H. angiospermum* [31], together with ethyl lithosprmate in the latter. 

Moreover, caffeic acid has very recently been identified in *H. ramosissimum* [5], while the taxonomic significance of neither RA nor caffeic acid in heliotropes and the Boraginacae plant family has been referred to and/or explained in the existing overview for heliotropes [5]. 

Within heliotropes, flavonoids are a common class of secondary metabolites, among which several express strong bioactivities. In the existing literature, several research surveys have characterized and reported various flavonoids, such as 5,4′-dihydroxy-7-methoxyflavanone, 4′-acetyl-5-hydroxy-7-methoxyflavanone, methoxy-3-[7′-methyl-3′-hydroxymethyl-2′,6′-octadienyl]phenol (in *H. glutinosum*), 5,3′-dihydroxy-7,4′-dimethoxyflavanone, 7-O-methyleriodictyol, 3-O-methylgalangin, filifolinol, filifolinyl senecionate (in *H. filifolium*), naringenin, filifolinoic acid, filifolinone, 3-oxo-2-arylbenzofuran (in *H. taltalense*, *H. sclerocarpum*), dihydroquercetine, quercetin (*H. strigosum*) [5]. 

In the present study, in *H. procubens*, the flavones luteolin-7-O-glucoside and trihydroxy-flavone hexoside, the flavanone naringenin-7-O-glucoside, as well as the flavonols kaempferol and quercetine-7-O-rutinoside (rutin) have been identified. Among them, luteolin-7-glucoside is a well-known metabolite isolated before from *H. tenellum* [32], but it is unfortunately not included in the phytochemical overview of *Heliotropium* genus [5]. Furthermore, luteolin-8-C-glucoside (orientin) has also been also reported for *H. ramosissimum* [16].

Additionally, naringenin-7-O-glucoside, a trivial flavonoid, has been previously identified in many heliotropes such as *H. taltalense* Phil., *H. sclerocarpum* Phil. [8], *H. sinuatum* [6,33], *H. glutinosum* Phil [7], *H. filifolium* and *H. huascoense* [5], together with the flavanone chrysin and sakuranetin in *H ellipticum* [34,35] 

Among flavonols, kaempferol and quercetin (the aglycon of rutin) and myricetin have been reported in *H. crispum* [36] so far.

In *H. crispum*, among other long chain carbon fatty acids, 5,8,12-trihydroxy-9-octadecenoic acid has been identified [17]. This chemical structure is commonly found in several plants of the Boraginaceae family as the basic molecule of octadecanoic acid (stearic acid), and different substitutions with hydroxyl groups have been determined in several plant species of the family, such as *Anchusa strigosa*, *Phyllocara aucheri, Cynoglottis barrelieri* [13].

There is a large demand from the market for natural extracts with antioxidant properties, especially from a plant origin, as many side effects caused by synthetic compounds have been reported to affect human health [37]. The high inhibition shown by the DPPH and ABTS assays is most likely related to the high phenolic content, as the identified phenolic compounds are extremely polar, containing multiple hydroxyl groups known for their antioxidative capacity. The mean values of the antioxidant potential of the extract in the ABTS assay were higher (68.31 ± 0.69 and 131.48 ± 6.66) compared to the DPPH assay (46.88 ± 0.64 and 93.43 ± 0.11). These results are explained by the difference in the mechanism of the radical interaction of these two assays, due to the fact that ABTS acts as both a hydrophobic and hydrophilic system, while DPPH acts only as a hydrophobic system. A connection between antioxidant activity and phenolic content has already been established for plants of the Boraginaceae family [12]. It is linked that much of the plant family’s biological effects (mostly antioxidant and anti-inflammatory) are relative to the concentration of phenolic compounds in such plants. 

Regarding the enzyme inhibition, it is noteworthy that the methanolic extract of the plant showed moderate inhibitory capacity against tyrosinase, an enzyme involved in skin whitening. 

## 4. Materials and Methods

### 4.1. Plant Material

The plant material was collected by Prof. M. Gupta and identified by M. Correa, Headmistress at the Herbarium of Panama University. The characteristics of the plant are plant number: 6476; family: Boraginaceae; name: *Heliotropium procumbens* Mill.; part of the plant: aerial part; collection habitat: Rio Grande; area: Cerca del Puente; amount: 1041 g. 

### 4.2. Extraction

The air-dried aerial parts of *H. procumbens* Mill. (900 g) were successively extracted in methanol (550 g) for 24 h at room temperature (3 × 0.3 L) and the rest (450 g) were extracted with water. The extracts were evaporated under reduced pressure to dryness, to afford 14.8 g of methanolic extract and 12.7 g of aqueous extract. 

### 4.3. Isolation of Phenolic Compounds

For the isolation of secondary metabolites, methanol extract (5 g) was fractioned on microcrystalline cellulose (20–160 μm, Merck) column chromatography with solvents (HPLC grade) of increasing polarity (saturated in H2O), gradient of cyclohexane(c-hex)/ethyl acetate (EtOAc) from 100:0 to 0:100, followed by EtOAc/MeOH (99:1 to 80:20) to yield 46 fractions (C1–C46). 

Fraction C8 (267 mg) was further subjected to column chromatography with Sephadex LH-20 (25–100 μm, Pharmacia) as the stationary phase and MeOH (HPLC grade) 100% as the mobile phase to isolate, and purify rosmarinic acid (21 mg), which was further identified through NMR (chloroform-*d*) and compared with bibliographic data [38]. The fractions C12 (48 mg) and C26 (100 mg) were lithospermic acid A and luteolin-7-O-β-D-glucoside, respectively. Both the compounds were structurally elucidated by NMR (chloroform-*d*) and a comparison of their spectroscopic data was made with those reported in the literature [39,40].

In order to evaluate the phenolic compounds through UHPLC–HRMS analysis, 1.9 g of the methanolic extract was subjected to chromatographic separation via open column chromatography with Sephadex LH-20 and MeOH as the mobile phase to give 55 fractions (HC.S1-HC.S55). The fractions HC.S8, HC.S10, HC.S17 and HC.S20, as well as the crude methanolic extract, were analyzed through UHPLC–HRMS analysis.

### 4.4. UHPLC–HRMS Analysis of Phenolic Compounds

Characterization of the phytochemical profile of methanolic extract was performed using ultra-high-performance liquid chromatography–mass spectrometry (UHPLC–HRMS). 

The UHPLC was performed employing a Vanquish UHPLC system (Thermo Scientific, Bremen, Germany) equipped with a binary pump, an autosampler, an online vacuum degasser, and a temperature-controlled column compartment. LC-MS grade methanol (MeOH) and formic acid (FA) were purchased from Fisher Scientific (Fisher Optima, UK) and LC-MS water was produced from a Barnstead MicroPure Water Purification System (Thermo Scientific, Germany). An Accucore Vanquish UPLC C18 (2.1 × 50 mm, 1.5 μm) reversed-phased column (Thermo Scientific, Germany) was used for the analysis. High-resolution mass spectrometry was performed on a Orbitrap Exactive Plus mass spectrometer (Thermo Scientific, Germany). Samples were injected at a concentration of 100 ppm diluted in MeOH/H2O 50:50. The mobile phase consisted of solvents A—aqueous 0.1% (*v*/*v*) FA, and B—acetonitrile. Different gradient elutions were performed for positive and negative ion mode detection and after optimization of the chromatography, the gradient applied was t = 0 min, 5% B; t = 3 min, 5% B; t = 21 min, 95% B; t = 26 min, 95% B. The flow rate was 0.3 mL/min and the injection volume was 5 μL. The column temperature was kept at 40 °C while the sample tray temperature was set at 10 °C. The ionization was performed at HESI, both in positive and negative mode. The conditions for HRMS for both negative and positive ionization modes were set as follows: capillary temperature, 320 °C; spray voltage, 2.7 kV; S-lense Rf level, 50 V; sheath gas flow, 40 arb. units; aux gas flow, 8 arb. units; aux. gas heater temperature, 50 °C. Analysis was performed using the Fourier transform mass spectrometry mode (FTMS) in the full scan ion mode, applying a resolution of 70,000, while the acquisition of mass spectra was performed in every case using the centroid mode. The data dependent acquisition capability was also used at 35,000 resolution, allowing for MS/MS fragmentation of the three most intense ions of every peak exceeding the predefined threshold, applying a 10 s dynamic exclusion. Normalized collision energy was set at 35. Data acquisition and analysis were undertaken by employing Xcalibur 2.1.

### 4.5. Total Phenolic Content (TPC)

The TPC of the extracts was calculated using the Folin–Ciocalteu method. Quantification was achieved by using a gallic acid reference curve. All measurements were conducted three times. Results were expressed as mg of gallic acid equivalents per gr of extract. UV/Vis absorption values at 765 nm were obtained using an Infinite M200 PRO TECAN reader (Tecan Group, Männedorf, Switzerland) [12].

### 4.6. Total Flavonoid Content (TFC)

The TFC of the extracts was calculated by means of an AlCl_3_ colorimetric assay. The results were obtained from a reference calibration curve of quercetin and expressed as mg of quercetin equivalents per g of extract. UV/Vis absorption values at 415 nm were obtained using an Infinite M200 PRO TECAN reader (Tecan Group, Männedorf, Switzerland) [12].

### 4.7. Radical Scavenging Activity

#### 4.7.1. 2,2′-Azino-bis(3-ethylbenzothiazoline-6-sulfonic acid) Diammonium Salt (ABTS) Assay

The free radical-scavenging activity of the extracts was determined by a ABTS radical cation (ABTS+) decolorization assay [12]. The sample absorbance was read at 734 nm. Trolox was used as a positive control and the radical scavenging activity of the tested extracts was expressed as mg of Trolox equivalents (mg TE/g extract). 

#### 4.7.2. 2,2-Diphenyl-1-picrylhydrazyl (DPPH) Assay

For the DPPH• assay, discoloration of the extract solution was measured to determine the antioxidant activity thereof [12]. The sample absorbance was read at 517 nm. Trolox was used as a positive control and the radical scavenging activity of the tested extracts was expressed as mg of Trolox equivalents (mg TE/g extract).

### 4.8. Reducing Power FRAP and CUPRAC Assays

FRAP (ferric reducing antioxidant power) and CUPRAC (cupric ion reducing antioxidant capacity) assays were used to determine the reductive activity of the extracts. Experimental procedures were undertaken as described previously [12] and reductive capability was expressed as mg of Trolox equivalents (mg TE/g extract) for both assays.

### 4.9. Total Antioxidant Capacity and Metal Chelating Activity

The total antioxidant activity of both extracts was evaluated by the phosphomolybdenum method [12]. Trolox was used as a positive control and the results were expressed as millimoles of Trolox equivalents (mmol TE/g extract). Metal chelating activity was determined, EDTA was used as a positive control and the results were expressed as milligrams of EDTA equivalents (mg EDTAE/g extract). 

### 4.10. Enzyme Inhibitory Activities

The ability of the methanol and aqueous extracts to inhibit the activity of selective enzymes was measured [12]. Cholinesterases (acetylcholinesterase (AChE) and butyrylcholinesterase (BChE)) inhibitory activity was expressed as mg of galanthamine equivalents (mg GALAE/g extract), tyrosinase inhibition as mg of kojic acid equivalents (mg KAE/g extract)**,** while for α-amylase and α-glucosidase the results were expressed as millimoles of acarbose equivalents (mmol ACAE/g extract). 

### 4.11. Expression of Results

All analyses were performed in triplicate. The results were expressed as mean values and standard deviation (SD). The differences in the extracts were investigated by using the Student *t*-test (*p* < 0.05) and this test was performed in Xlstat 2018.

## 5. Conclusions

The presented study is the first record on *Heliotropium procumbens* and it is focused on the determination of the phenolic profile of the plant as well as the evaluation of in vitro antioxidative, neuroprotective and antidiabetic effects. It is noteworthy that the taxonomic significance of the reported phenolic secondary metabolites and especially the caffeic derivatives (mostly caffeic acid and RA) has been proved, as well as the most common flavonoids, and their potential contribution to the expressed bioactivities. The importance of exploiting natural antioxidants of herbal origin has increased significantly at present due to the fact that synthetic antioxidants have been suspected to express adverse reactions to human health [14]. According to the results, *Heliotropium procumbens* showed a rich phenolic profile, which is related to its strong antioxidant activity as determined by TPC, DPPH and ABTS assays, as well as moderate tyrosinase inhibition, which is also of high interest towards required systematic procedures necessary to rich extracts in various active secondary metabolites for further safe exploitation.

## Figures and Tables

**Figure 1 molecules-28-01008-f001:**
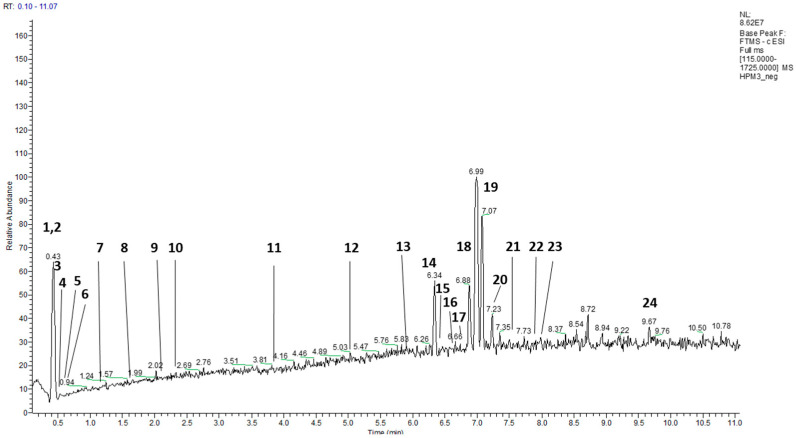
Characterization of the phenolic constituents of *Heliotropium procubens* extract by UHPLC–MS.

**Table 1 molecules-28-01008-t001:** Secondary metabolites identified from methanolic extract and fractions through UHPLC–HRMS.

No	Retention Time	Identification	Chemical Structure	*m*/*z*	Ion Mode	MS2	Ref.
**1**	0.39	bis-hexoses	C_12_H_22_O_11_	341.1097	[M-H]^−^		[15]
**2**	0.44	malic acid	C_4_H_6_O_5_	133.0133	[Μ-H]^−^	115, 71	[16]
**3**	0.46	isobergapten	C_12_H_8_O_4_	215.0327	[M-H]^−^	125, 157	[17]
**4**	0.77	dimethoxy-hydroxybenzoic acid (syringic acid)	C_9_H_10_O_5_	197.0452	[M-H]^−^	179, 142, 135, 123, 114	[18,19,20]
**5**	0.8	dihydroxybenzoic acid	C_7_H_6_O_4_	153.0186	[M-H]^−^	109	[16]
**6**	0.94	cοumarate glucoside	C_20_H_16_O_8_	321.1020	[M-H_2_O-H]^−^	97	[21]
**7**	1.22	hydroxybenzoic acid	C_7_H_6_O_3_	137.0235	[M-H]^−^	109, 93	[16]
**8**	1.65	dihydroxy methoxy benzoic acid	C_8_H_8_O_5_	183.0294	[M-H]^−^	168, 141, 111	[21]
**9**	2.10	caffeic acid	C_9_H_8_O_4_	179.0345	[M-H]^−^	179, 161, 135, 124	[16]
**10**	2.37	vanilic acid glucoside	C_14_H_18_O_9_	329.0919	[M-H]^−^	97	[21]
**11**	3.97	caffeoyl hexoside	C_15_H_18_O_9_	341.0917	[Μ-H]^−^	241, 150, 97	[21]
**12**	5.04	cοumaric acid derivative	C_15_H_16_O_4_	305.1067	[M+FA-H]^−^	97	[21]
**13**	5.99	trihydroxyflavone dihexoside (rutin)	C_27_H_29_O_16_	609.1475	[Μ-H]^−^	285	[22]
**14**	6.36	luteolin-7-O-glucoside *	C_21_H_20_O_11_	447.0941	[M-H]^−^	285, 151, 96	[23]
**15**	6.47	naringenin-7-O-glucoside	C_21_H_22_O_10_	479.1204	[M+FA-H]^−^	167, 153	[24]
**16**	6.63	methyl-catechin	C_16_H_16_O_6_	303.0914	[Μ-H]^−^	97	[25]
**17**	6.76	ferulic acid (or isomer)	C_10_H_9_O_4_	193.0502	[Μ-H]^−^	178, 161, 137	[17]
**18**	6.88	trihydroxy-flavone hexoside	C_21_H_20_O_10_	431.0989	[M-H]^−^	268, 152, 109	[24]
**19**	7.04	rosmarinic acid *	C_18_H_16_O_8_	359.0780	[M-H]^−^	197, 179, 161, 135, 72	[26]
**20**	7.23	lithospermic acid *	C_27_H_22_O_12_	537.1051	[Μ-H]^−^	295, 185, 109	[17,21]
**21**	7.53	salvianolic acid B	C_36_H_30_O_16_	717.1485	[Μ-H]^−^	609, 536, 362, 321, 279, 185, 109	[17,25]
**22**	7.89	rosmarinic acid methyl ester	C_19_H_18_O_8_	373.0935	[M-H]^−^	197, 175, 171, 135, 72	[26]
**23**	7.92	kaempferol	C_15_H_10_O_6_	285.0411	[M-H]^−^	267, 223, 189, 174	[25]
**24**	9.68	5,8,12-trihydroxy-9-octadecenoic-acid	C_18_H_34_O_5_	329.2339	[M-H]^−^	211, 171, 139, 99	[25]

* Isolated compounds.

**Table 2 molecules-28-01008-t002:** Antioxidant properties of *H. procumbens*.

Extracts	TPC (mg GAE/g Extract)	TFC (mg RE/g Extract)	DPPH (mg TE/g Extract)	ABTS (mg TE/g Extract)	CUPRAC (mg TE/g Extract)	FRAP (mg TE/g Extract)	Metal Chelating (mg EDTAE/g Extract)	Phosphomolybdenum (mmol TE/g Extract)
ME	32.20 ± 0.22	0.70 ± 0.15	46.88 ± 0.64	68.31 ± 0.69	166.06 ± 4.29	79.17 ± 0.62	11.76 ± 1.12	1.74 ± 0.06
WE	53.47 ± 0.52	12.39 ± 0.70	93.43 ± 0.11	131.48 ± 6.66	332.97 ± 3.52	167.58 ± 5.47	13.97 ± 0.06	1.81 ± 0.03

Values expressed are means ± S.D. of three parallel measurements. GAE: gallic acid equivalent; RE: rutin equivalent; TE: Trolox equivalent; EDTAE: EDTA equivalent.

**Table 3 molecules-28-01008-t003:** Enzyme inhibitory effects of *H. procumbens*.

Extracts	AChE Inhibition (mg GALAE/g Extract)	BChE Inhibition (mg GALAE/g Extract)	Tyrosinase Inhibition (mg KAE/g Extract)	Amylase Inhibition (mmol ACAE/g Extract)	Glucosidase Inhibition (mmol ACAE/g Extract)
ME	na	1.17 ± 0.25	25.05 ± 0.17	0.22 ± 0.01	1.97 ± 0.14
WE	0.48 ± 0.04	na	8.82 ± 0.80	0.07 ± 0.01	2.08 ± 0.01

Values expressed are means ± S.D. of three parallel measurements. GALAE: Galantamine equivalent; KAE: kojic acid equivalent; ACAE: acarbose equivalent; na: not active.

## Data Availability

Not applicable.

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
