# Peer review of "Heliotropium procubens Mill: Taxonomic Significance and Characterization of Phenolic Compounds via UHPLC–HRMS- In Vitro Antioxidant and Enzyme Inhibitory Activities"

_molecules, 2023, doi:10.3390/molecules28031008_

Round 1

Reviewer 1 Report

Manuscript Number: molecules-2138940
entitled: Heliotropium procubens Mill: Taxonomic significance and characterization of phenolic compounds via UHPLC-HRMS- In vitro antioxidant and enzyme inhibitory activities

This is an interesting scientific study. Therefore, the manuscript is suitable for Molecules after considering the below comments:

  1. Would you please show the structures of the main secondary metabolites mentioned in the manuscript?
  2. Table 1. All secondary metabolites should be described using lowercase, not capital letters, such as “Dihydroxybenzoic acid.”
  3. Plant material. It possibly will be nice to present the main photos of plants mentioned in the manuscript.
  4. Conclusion part. Please use the whole name of H. procumbens.

Author Response

This is an interesting scientific study. Therefore, the manuscript is suitable for Molecules after considering the below comments:

  1. Would you please show the structures of the main secondary metabolites mentioned in the manuscript?
  • The structures of the isolated secondary metabolites have been introduced in the Supplementary file.
  1. Table 1. All secondary metabolites should be described using lowercase, not capital letters, such as “Dihydroxybenzoic acid.”
  • Lowercase letters have been used for all secondary metabolites in Table
  1. Plant material. It possibly will be nice to present the main photos of plants mentioned in the manuscript.
  • A photo of the aerial parts of the studied plant has been added in the Supplementary file
  1. Conclusion part. Please use the whole name of H. procumbens.
  • The whole name Heliotropium procumbens has been used in the Conclusion part instead of the short procumbens.

Reviewer 2 Report

Structures of the compounds were elucidated using mass spectra and 3 isolated compounds' structures were confirmed by NMR.

# I suggest authors to insert a scheme (s) of those compounds which were structurally determined by mass with fragmentation pattern (Structure to fragments). please see the published articles

1. (supporting file of Separations 20229(12), 400; https://doi.org/10.3390/separations9120400

2. Molecules 202025(24), 5903; https://doi.org/10.3390/molecules25245903

# UPLC spectra should be inserted as a figure and indication should be given for all the compounds with numbering. and retention times.    

Author Response

Structures of the compounds were elucidated using mass spectra and 3 isolated compounds' structures were confirmed by NMR.

# I suggest authors to insert a scheme (s) of those compounds which were structurally determined by mass with fragmentation pattern (Structure to fragments). please see the published articles

1.supporting file of Separations 2022, 9(12),400; https://doi.org/10.3390/separations9120400

  1. Molecules2020, 25(24), 5903; https://doi.org/10.3390/molecules25245903

- Structures of the compounds, which were isolated and structurally determined, with fragmentation pattern, have been added in the Supplementary file.

# UPLC spectra should be inserted as a figure and indication should be given for all the compounds with numbering. and retention times.

  • The mass spectra have been inserted as figures and indication with numbering and retention times has been given for all the determined compounds in the Supplementary file.

Round 2

Reviewer 2 Report

The manuscript is now acceptable for publication.